# Submerged Vane Technology in Colombia: Five Representative Projects

**Carlos Rodríguez-Amaya ***, **Argelino Durán-Ariza *** **and Santiago Duarte-Méndez ***

Hidroconsulta SAS, Colombian Society of Engineers, SCI, Bogotá 48 #9175, Colombia
* Correspondence: tecnica@hidroconsulta.com (C.R.-A.); gerencia@hidroconsulta.com (A.D.-A.); sduartem@hidroconsulta.com (S.D.-M.)

**Abstract:** An innovative research-based technology has been applied for the first time in Colombia to improve the navigability of the Magdalena river in a zone of the city of Barrancabermeja, Department of Santander. The result of installation of the submerged vane technology demonstrated its effectiveness in sediment management and motivated its further use as a solution to problems of erosion, scour and meander evolution, which are common occurrence in the rivers of the country. Since this 1991 installation, more than 18 projects have been completed and the technical effectiveness of the system has been improved. Compared to traditional solutions, the results demonstrate beneficial economic impacts due to shorter execution times, reduction in annual maintenance costs, and diminished environmental impacts. Characteristics of design and construction and results obtained from five projects are described that are representative of the diversity of conditions and difficulties for the application of this technology in Colombia. Lessons learned for adaptation by river management authorities are derived from the study.

**Keywords:** submerged vanes; sediment management; river evolution; rivers of Colombia

---

## 1. Introduction

It has been well established in the technical literature that sediment transport in rivers and its associated processes of undermining, aggradation, bank erosion and meandering are the most important problems of river hydraulics. Their control and management are indispensable to establishing conditions for navigation, ensuring the supply of water for various uses and preventing the erosion and undermining of bridges, docks and riverbanks, among other structures.

This paper addresses the effectiveness of a river training methodology called submerged vanes, which was developed at the University of Iowa Institute of Hydraulic Research (IIHR) and reported through studies by Odgaard and Kennedy (1983) [1]; Odgaard and Spoljaric (1986) [2]; and Odgaard and Wang (1991a) [3]. The question addressed here is how well the technology works in the specific river conditions of Colombia, where river training is necessary due to a large number of nearby roads and the high risks of frequent destruction. Submerged vanes offer substantial improvements over the commonly adopted solution, which is based on methods involving high costs and the requisition of surrounding lands. This technology also offers advantages for river stabilization that could facilitate navigation.

Traditional methods applied in Colombia are: (1) gabions (curled or boarded), for direct protection of banks; (2) directional spurs to promote sedimentation and push erosive flow towards the center of a channel; (3) dredging bed material; and (4) closing secondary channels with dredged materials from the river or using special structures. Each of these methods has its own specific challenges, but in general all are becoming increasingly difficult to apply due to high costs and environmental restrictions.

This article reports on the effectiveness of some 18 projects in the country, which were designed using laboratory results reported by Odgaard and Kennedy (1983) [1]; Odgaard and Spoljaric (1986) [2]; and Odgaard and Wang (1991a) [3], as well as results of the first applications reported in Odgaard and Mosconi (1987) [4] and Odgaard, and Wang (1991b) [5]. The projects are shown in Figure 1.

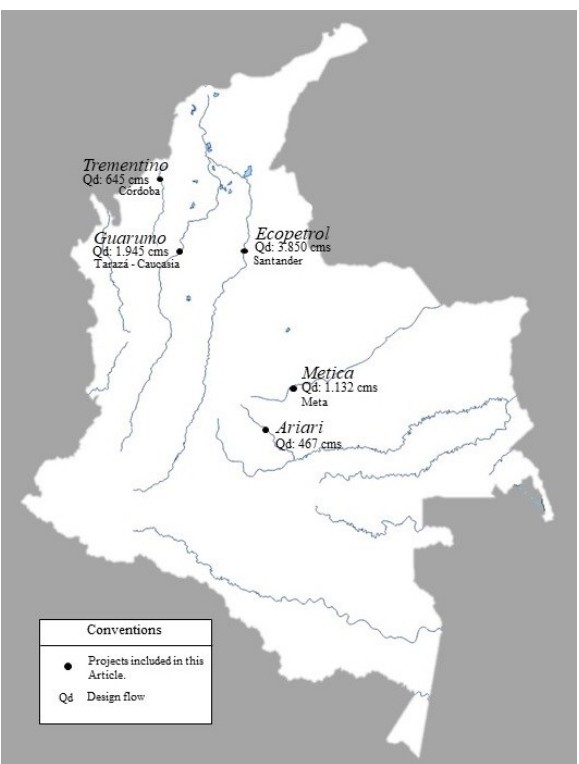

**Figure 1.** Location of the five projects with submerged vanes mentioned in this article.

The positive results obtained with this technology in several rivers of the country include: short term construction, rapid sedimentation and, most importantly, notable savings to the national economy. In this study, five representative projects are described in some detail with reference to design, construction and results.

## 2. Technical Basis for the Use of Submerged Vanes

The submerged vane technology developed by the IIHR uses the effects of submerged vanes on the generation of secondary forces, which alter the magnitude and direction of the bed shear stresses and cause a change in the distribution of velocity, depth and sediment transport in the area influenced by the vanes (Odgaard, 2009) [6]. Based on these effects, criteria for the design of submerged vane fields were established and applied for river and bank stabilization and control of the undermining of bridges (Odgaard and Mosconi, 1987 [4]; Odgaard, and Wang, 1991b [5]) as well as for sediment control in catchments (Wang et al., 1996 [7]). The resulting design criteria used in Colombia were positioning angle and dimensions of the vanes, number of vanes per set and number of sets, separation between vanes and arrays, and distance to the riverbank. In some cases, variations were needed due to local conditions.

The hydraulic design of a submerged vane field includes the layout, the location coordinates of the fastening piles, details about distances to the river bank, and the separation between the vanes and system arrays. The structural design is carried out for the manufacture of the vanes, the selection of the piles and the characteristics of the metal rings fixing the piles, from which the required plans are drawn.

Greater efficiency is achieved with the "Iowa vanes" of improved geometry (Odgaard, 2009 [6]), which are fixed on the riverbed with a single metal pilot with an 'H' profile. For the practical matters of installation and costs, the projects usually have thin flat prefabricated plates of solid or lightened concrete, or metal lattices coated with high-density polypropylene sheets, depending the local conditions. Two piles are used for fix the vanes in place, one at each end of the vane, allowing the vane to slide vertically if unexpected changes in river dynamics modify the angle of attack of the flow.

The installation is from a barge using equipment for lifting and transferring vanes and piles and driving the piles into the bed. Topographic control uses grounded equipment where the coordinates and alignments of the piles are verified. The installation is important for the success of the systems, and the procedure is the same but adapted to local conditions according to availability of equipment, flow conditions, depth and speed of flow, and type and weight of panel vanes. Sometimes there are constraints to not restrict the local transport of passengers and products. When no barges are available or the river is not navigable, floating structures are used that are transported via truck.

In the next section, specific features of five projects are discussed in greater detail.

## 3. Applications of Submerged Vane Technology in Colombia

### 3.1. Magdalena River: ECOPETROL Aromatics Pier in Barrancabermeja (1991)

This was the first submerged vane project designed and built in Colombia, with the aim of stabilizing the Magdalena river bed in front of the Aromatics pier of the Colombian Petroleum Company ECOPETROL [8], where sedimentation caused by the river had restricted access to the barges used in the transport of crude oil for export, resulting in environmental risks (due to possible product spills) and high operating costs.

The objective of the project was to eliminate the continuous dredging work in front of the dock so as to ensure access for vessels of 3.05 m of draft. The vanes were to be located approximately 250 m from the dock. The annual cost of dredging was, according to ECOPETROL's report, USD 1.15 million, a cost that was desirable to eliminate. On the other hand, because dredging did not produce stable results, in the short term new and frequent dredging was required.

The construction of a field of submerged vanes was proposed to establish a wide berm along the left bank in front of the pier which would lower the thalweg by undermining the level of the river bed and the approach to the pier, thus promoting the natural dredging of sediments (Hidroconsulta, 1991) [8].

The minimum water level in front of the pier, measured in 1936, was 70.09 m above sea level. The river bed at the time of the design was 66 m above sea level, so navigational possibilities and loading ties for ships with a required draft of 3.05 m at approximately 250 m from the pier were limited. There was a maximum flow rate of 5500 $m^3$/s and water depth had risen more than 9.20 m.

The conditions for the design in cross section 9-9 (Figure 2) were as follows: flow, Q = 3.850 $m^3$/s; average slope, S = 0.00029 m/m; mean sediment diameter, $D_{50}$ = 4.69 mm; width of the full section, B = 219.77 m; speed range, $V_{max}$ = 2.29 m/s and $V_{min}$ = 1.69 m; radius of curvature, $R_c$ = 770 m' approximately. For the design, vanes of 1.50 x 4.50 m were adopted, with a separation of 4.0 m between panels and 40 m between arrays (Odgaard and Wang, 1991a [3]; Hidroconsulta, 1991 [8]). According to Odgaard and Mosconi (1987) [4], the resistance parameter of the channel (m) was 7.29 m, corresponding to a Darcy-Weisbach friction coefficient of 0.025. It was concluded that there was a need for at least 58 vanes to be distributed in nine sets of seven units each. However, due to the lack of a clear definition of the edge for section 7-7 (Figure 3), it was decided that the first array would comprise 15 vanes, while the other eight would consist of seven units each, for a final design with a total of 71 vanes (Figure 3).

The panels were made of reinforced concrete of f'c' = 4000 psi (280 $Kg/cm^2$), 0.20 m thick, with four rings in a plate of 1/4" × 0.10 m as a guide for the metal piles for which steel pipes 14" × 7 m in length were used fixing them in the bed. The initial design envisioned a 10" pipeline, but the use of a 14" oil pipeline, supplied directly by ECOPETROL, was chosen. Figure 4 shows the storage of vanes at the riverfront and the installation process.

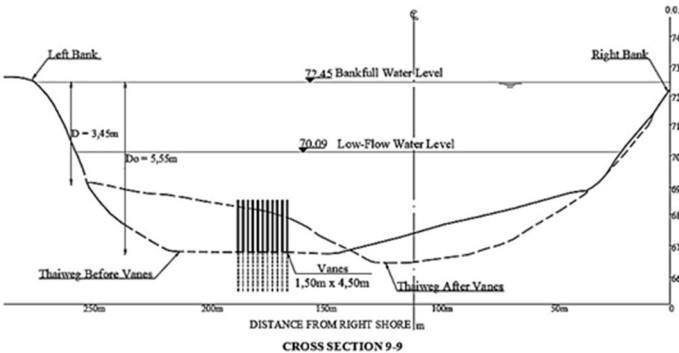

**Figure 2.** Cross section 9-9 used in the design of the vane field in front of the Aromatics dock.

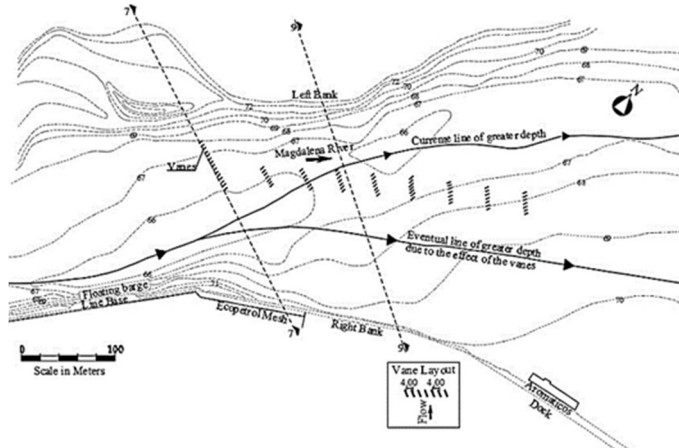

**Figure 3.** Location of the submerged vane field in front of ECOPETROL's Aromatics pier (Hidroconsulta, 1991).

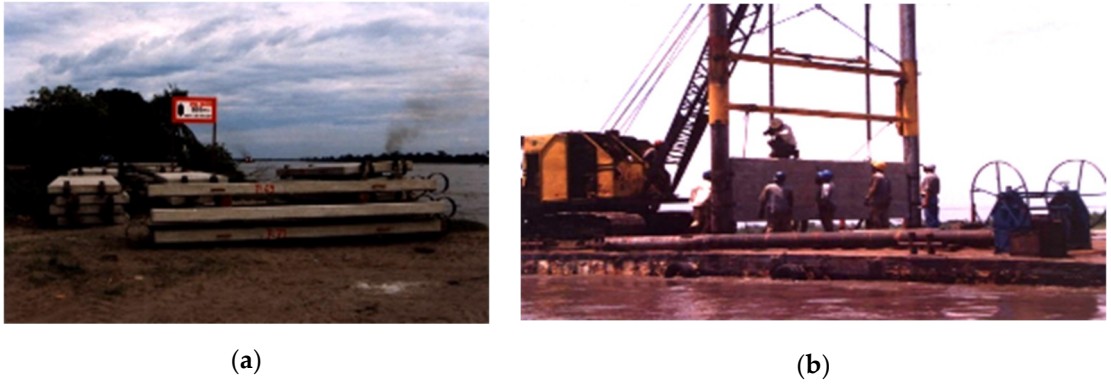

|     |     |
| --- | --- |
| (**a**) | (**b**) |

**Figure 4.** Magdalena River, ECOPETROL Aromatics pier: (**a**) storage of vanes in the riverfront before installation; (**b**) vane installation process (Hidroconsulta, 1993).

By facilitating the access of ships to the dock 45 days after it was installed, the scouring generated by the vane system near the pier was, in practice, immediate. The modification of the cross-section through sedimentation around the vanes and the deepening of the canal in front of the pier are shown in Figure 5 for the years 1991–1997, a period during which the system was followed. The figure also includes the results of a bathymetry performed 20 years later in 2012, which highlights a new sedimentation progression near the pier, the causes of which have not been investigated but suggest that the design and construction of a second stage of vanes should be considered.

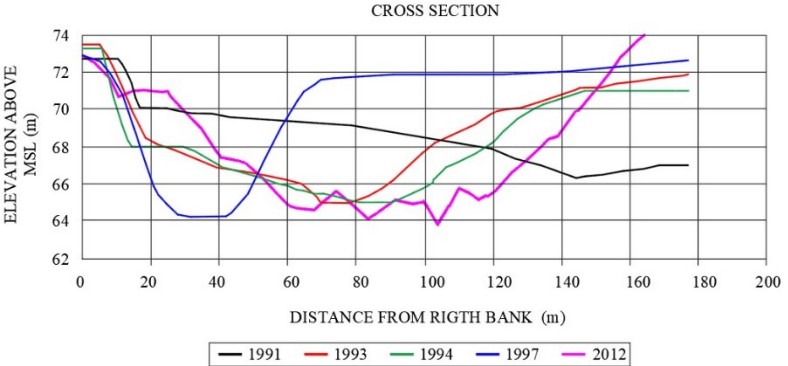

**Figure 5.** Effect of the submerged vane field installed in front of ECOPETROL's Aromatics pier in 1993 over time.

This project required consideration of local conditions during installation. A rigid metal structure was designed and manufactured to assemble the piles and vane on the barge, and was then moved to the edge of the barge where it was temporarily tied down with cables. Once the location and alignment were verified, the piles were slipped and partially driven, and the panel vane was untied so that it would fall. Local transport was required and each vane had to be installed in the shortest possible time. A similar procedure is currently applied when depths are greater than 9 m.

Success was shown by this first work in Colombia using submerged vanes, and the process was executed in just three and a half months, between 1 June and 23 August, 1993, at a cost of USD 218,750. The expenditure of US 1.15 million/year was abolished for a period of approximately 10 years, with significant savings to the Colombian state.

### 3.2. Sinú River, in Trementino (2004)

The Sinú River in the Trementino (Project nominated for the 2005 National Engineering Award, of the Colombian Society of Engineers) sector has a large meander with a progressive evolution of approximately 3 m/year, as determined for the period 1989–2004 (Hidroconsulta [9]). As a result, half a bench of the San Bernardo del Viento-Lorica road (a length of approximately 160 m) was destroyed, affecting its safety and the local environment. After this, the meander was protected with mangrove piles and the riverbank was stabilized by placing hexapods at its base, but without positive results.

In 1998, five spurs were constructed using concrete-filled bags to protect 240 m of the bank at a cost of USD 605,000, but the expected effects were not achieved. Subsequently, three new spurs were added in 2004 which were spaced to 60 m, consisting of two lines of steel pipe piles spaced at 1 m and filled with mortar in polypropylene bags. These new spurs and remnants of the previous solutions (mangrove piles and hexapods) were able to protect 320 m of the riverbank, at a total cost of USD 1,068,000.

As an alternative solution, and in order to definitively counteract the recurring problem of erosion and undermining in Trementino, a field of submerged vanes was designed that was able to protect 630 m of riverbank. The system features are described below. Figure 6 presents the initial conditions of the project site, the representative geometry of the river, and the three sectors in which the meander was subdivided for design purposes, which are compared with the typical parameters of each curve and the general distribution of the various arrays of vanes that formed the submerged vane field.

The design was carried out under the following conditions: flow to full bench, 645 m$^3$/s; water temperature, 25 °C; slope of the energy line, $S_0 = 1.74 \times 10^{-4}$ m/m. In general, the depth of flow is high in the Trementino sector, in some cases being more than 15 m (Hidroconsulta, 2004 [10]).

To define the options for vanes that are most suited to project objectives, an initial sensitivity analysis of possibilities was carried out using the BVANE model (Odgaard, 1996 [10]), indicating the convenience of using panels of 3.0 × 6.0 m with incidence angles of 20°, in sets of three and four vanes located between 4.0 and 5.0 m apart, depending on the sector. The final design is presented in Table 1.

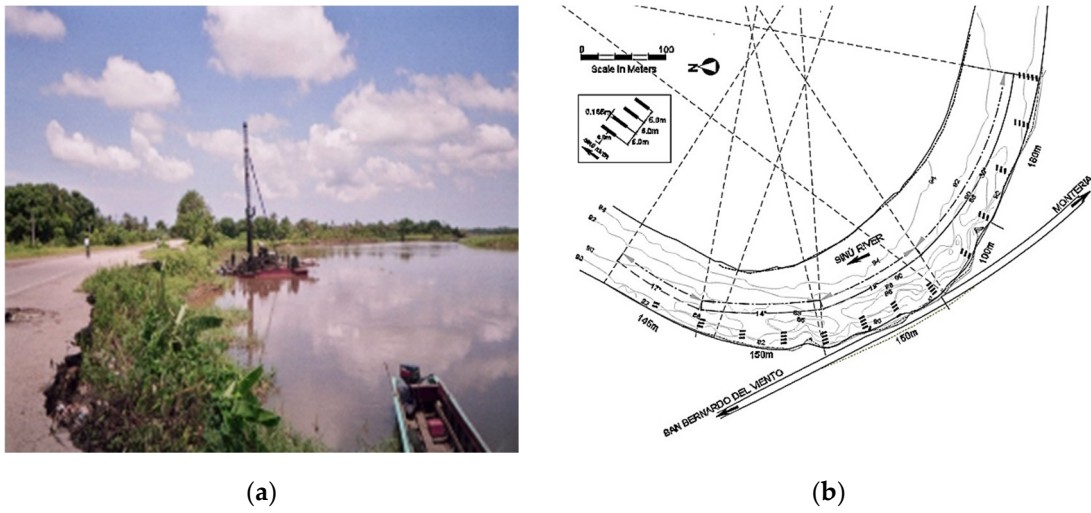

(**a**)　　　　　　　　　　　　　　　　　　　　　　　　　(**b**)

**Figure 6.** Sinú River, Trementino: (**a**) condition at the start of the project; (**b**) representative geometry used in the design (Hidroconsulta, 2004).

**Table 1.** Sinú River, Trementino: design of the submerged vane field (Hidroconsulta, 2004 [10]).

| Sector | Dimensions H (m) × L (m) × e(m) | Panels/Set (Number) | Separation Panels (m)/Array (m) | Protected Length (m) | Arrays (No.) | Total Vanes |
|---|---|---|---|---|---|---|
| 1A | 3.0 × 6.0 × 0.165 | 3 | 5.0/60.0 | 180.0 | 4 | 15 |
| 1B | 3.0 × 6.0 × 0.165 | 3 | 5.0/50.0 | 100.0 | 1 | 3 |
| 2 | 3.0 × 6.0 × 0.165 | 4 | 4.0/50.0 | 150.0 | 4 | 16 |
| 3 | 3.0 × 6.0 × 0.165 | 4 | 4.0/50.0 | 200.0 | 3 | 12 |
| | | | TOTAL | 630.0 | 12 | 46 |

The vanes were initially designed using reinforced concrete, and were 6.0 × 3.0 × 0.165 m, weighing 7.13 tons each; however, they were later redesigned to facilitate handling and installation operations. As a result, metal lattices coated with sheets of high-density polyethylene were made reducing the individual weights to only 780 Kg. The installation scheme and dimensions of the vanes that were ultimately adopted are presented in Figure 7.

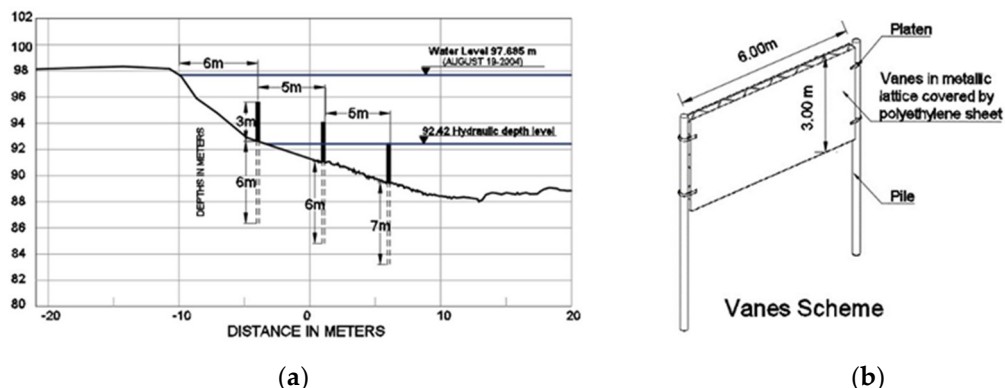

(**a**)　　　　　　　　　　　　　　　　　　　　　　　　　(**b**)

**Figure 7.** Sinú River, Trementino: (**a**) installation scheme; (**b**) characteristics of the panels (Hidroconsulta, 2004 [9]).

The installation was carried out over 85 effective days between mid-September and December 2004 under extreme flow and water level conditions (Figure 8), as the flow rates during the construction

stage were above the annual average. Due to the local river dynamics during the construction it was necessary to relocate some arrays and increase their number to 13, without increasing the total of 46 vanes.

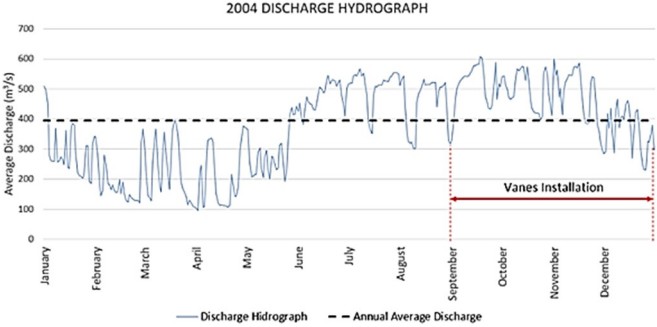

**Figure 8.** Sinú River, Trementino: daily flow series in 2004 (Source: Limnimetric Station The Doctrine, IDEAM).

The effectiveness of the vane field installed in Trementino was determined by observing the changes in sedimentation and the transfer of the thalweg to the center of the channel over the course of a few months (Figure 9). These changes facilitated the rehabilitation and resurfacing of the affected section of the road. No maintenance has been required to date (2019).

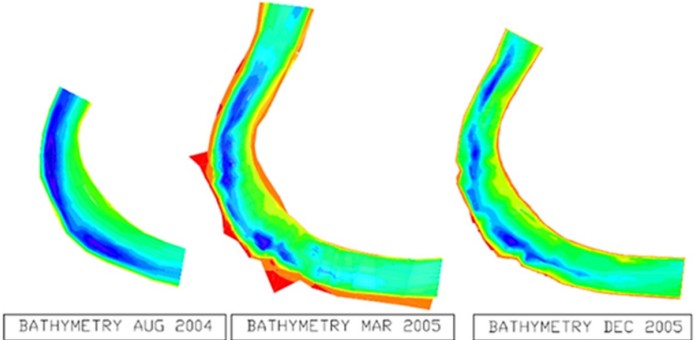

**Figure 9.** Sinú River, Trementino: evidence of the almost immediate effect of the displacement of the thalweg through the submerged panel field installed.

Figures 10 and 11 show the long-term effects (in the years 2006, 2011, and 2018) of the vane field installed in 2003. To date (2019), the site is stable, and no maintenance has been required since its construction.

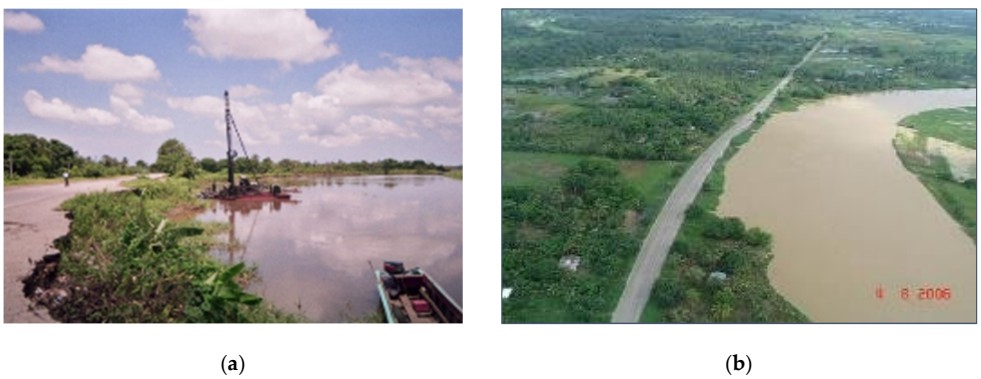

(**a**)　　　　　　　　　　　　　　　　　　　　　　　　　　　　　(**b**)

**Figure 10.** Sinú River, Trementino: (**a**) at the beginning of the panel installation in September 2003; (**b**) condition in August 2006, three years later (Hidroconsulta, 2003; 2006).

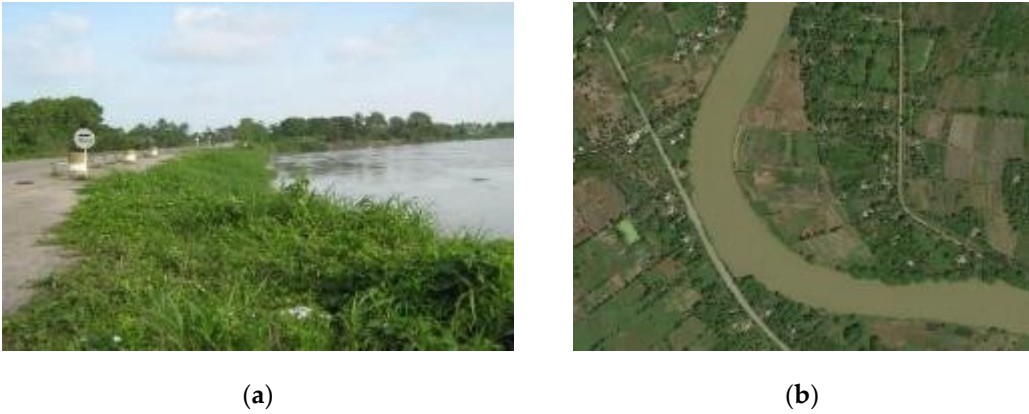

(**a**)　　　　　　　　　　　　　　　　　(**b**)

**Figure 11.** Sinú River, Trementino: (**a**) condition in March 2011, seven and a half years after installation; (**b**) condition in 2018, 13 years after installation (Hidroconsulta, 2011; ESRI, 2018 [11]).

### 3.3. Rio Metica, in Puerto López, Sector 2 (2007)

The objective of the project was to control the evolution of a huge meander on the left bank of the Metica River in Puerto López through the so-called Phase 2 (Figure 12), which has approached the Puerto López–Puerto Gaitán highway at a rate of 39 m/year between 1939 and 2007 [12]. The road was less than 50 m from the river in 2005 (Figure 12a).

A previous intervention consisted of six spurs built with two lines of metal pipe with the space between them filled with bags of concrete. Except for two of them (Figure 13), the spurs were soon destroyed by the river. Thus, it was decided to apply the submerged vane tecnology based on the geometry shown in Figure 12b (Hidroconsulta, 2005 [13]), which was adjusted two years later (Hidroconsulta, 2007 [12]) when its construction was initiated.

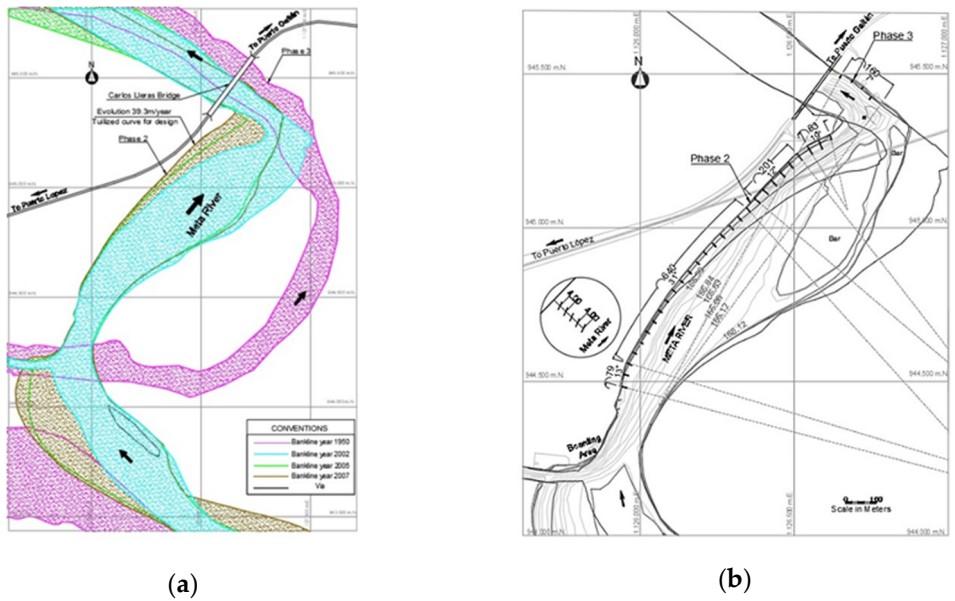

(**a**)　　　　　　　　　　　　　　　　　(**b**)

**Figure 12.** Rio Metica, Puerto López: (**a**) summary of the analysis of the evolution of the river among 1950, 2002, 2005, and 2007; (**b**) sector geometry and panel field design. Note the proximity of the river to the road in 2005 (Hidroconsulta, 2005 [13]; 2007 [12]).

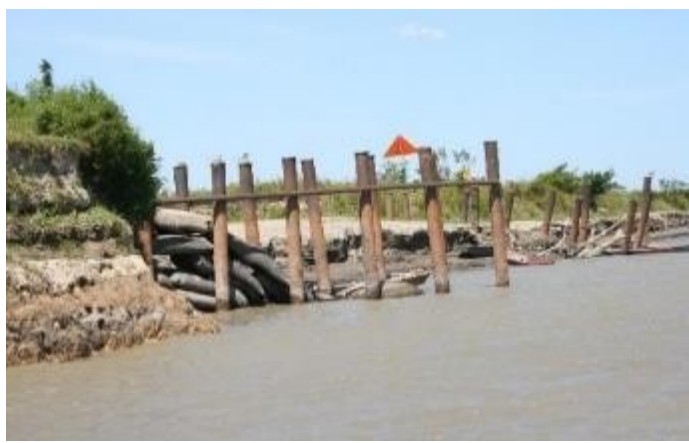

**Figure 13.** Rio Metica, Puerto López: the two spurs that remain, although highly deteriorated, of the six previously built.

For the estimated discharge of 1132 m$^3$/s that fills the section, the results of the HEC-RAS model (USACE, 2001 [14]) were used in model BVANE (Odgaard [10]) (Figure 14) to design the vane field for units of 1.50 × 4.50 m. The general parameters used were as follows:

| | |
|---|---|
| Water temperature, (T °C) | : 20 °C |
| Average particle size, D$_{50}$, mm | : 0.08 |
| Flow to full bench, m$^3$/s | : 1.132 |
| Water level slope, S0 (m/m) | : 1.74 × 4.40 |
| Incidence angle | : 10°, 15°, 18° or 20° |
| Number of panels per set | : 4, 5, 6 or 7 |
| Maximum separation between sets, m | : 45 |
| Separation between panels, m | : 3 and 4, 5 |

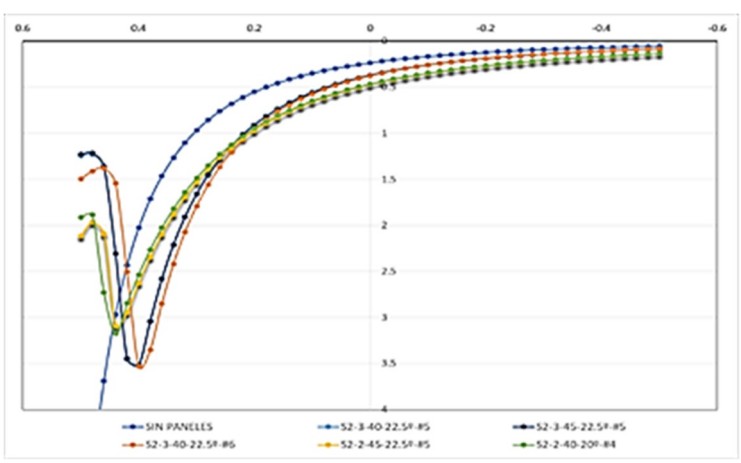

**Figure 14.** Rio Metica, Puerto López: optimal selection for the vane system.

A submerged vane field was established to protect 1048 m of riverbank that consisted of 134 vanes distributed in 27 arrays with 40 or 45 m between them and 4.0 or 4.5 between individual vanes, depending on their location. In the redesign (Table 2), 1.6 × 4.4 × 0.2 m vanes were adopted using reinforced concrete lightened with expanded polystyrene sheets, which attained a good general performance and improved the structural conditions for its manufacture, handling, and installation. Steel piles of 6" × 7.0 m lengths were used.

**Table 2.** Redesign of the submerged vane system for the Metica River in Puerto López, Department of Meta (Hidroconsulta [12]).

| Sector | Curve | Number of Vanes per Array | Separation (m) | | Protected Bank Length (m) | Number of Arrays | Total Number of Vanes |
|---|---|---|---|---|---|---|---|
| | | | Between Vanes | Between Arrays | | | |
| 3 | 0 | 5 | 4.0 | 40.0 | 79.2 | 2 | 10 |
| 3 | 1 | 4 o 5 | 4.0 | 40.0 | 653.7 | 17 | 74 |
| 3 | 2 | 6 | 4.5 | 45.0 | 249.6 | 6 | 36 |
| 3 | 3 | 7 | 4.5 | 45.0 | 65.3 | 2 | 14 |
| | | | | TOTAL | 1047.8 | 27 | 134 |

The system response in terms of the sedimentation produced was obtained in a very short time, as can be seen in Figure 15a,b.

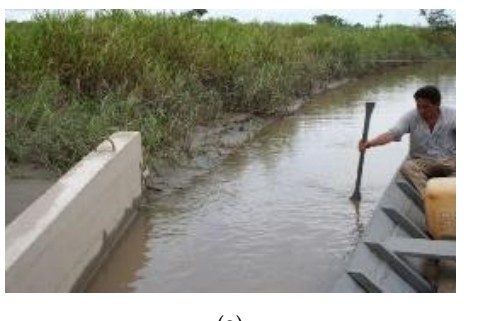
(**a**)

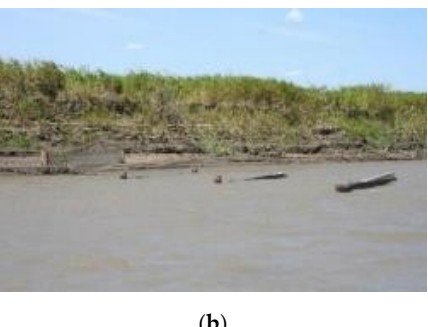
(**b**)

**Figure 15.** Metica River, Puerto López. (**a**) The sedimentation around the first panels of one of the sets generated bed depths of approximately 0.50 m, 15 days after installation. (**b**) Sedimentation progression as a result of submerged panels in January 2008 (Source: Hidroconsulta, 2007).

The cost of the project amounted to USD 448,900, at the rate of USD 429 per meter of protected bank, while the conventional spur solution cost USD 1,887,900, meaning that a savings of USD 1,439,000 was generated.

*3.4. Cauca River, Guarumo Site (2009)*

The project was located in the department of Antioquia, where the Tarazá-Guarumo-Caucasia highway is parallel to the left bank of the Cauca River [15] (Project awarded the Honorable Mention of the National Engineering Award 2010, of the Colombian Society of Engineers). During the decade from 1970 to 1980, this sector demonstrated problems of erosion and lateral undermining that continued and increased along 1900 m, posing a serious threat to the stability of the road. To protect the most affected part, a direct bank protection was built in 2004, which was 40 m in length with two directional spurs at its ends that were 15 and 11 m long, respectively (Figure 16). This was an initial measure for a plan to include six spurs and protect up to 515 m of shore at an estimated cost of US 1.8 million, plus the cost of recommended dredging.

To reduce the high cost, two submerged vane fields were designed and constructed to protect 616 m of shore in two stages: the first of 352 m, and the second of 264 m, with two subsectors each, based on local geometric conditions (Figure 17). The cost of these two stages amounted to USD 725,400, resulting in savings of USD 1.07 million for the spurs and dredging solution combined.

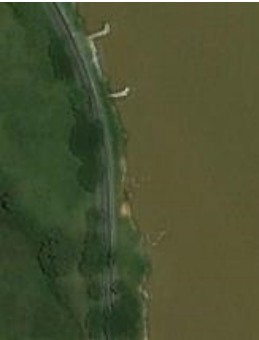

**Figure 16.** Cauca River, Guarumo: directional spurs around the 40 m long defense as a left bank protection (Source: BING Maps Aerial).

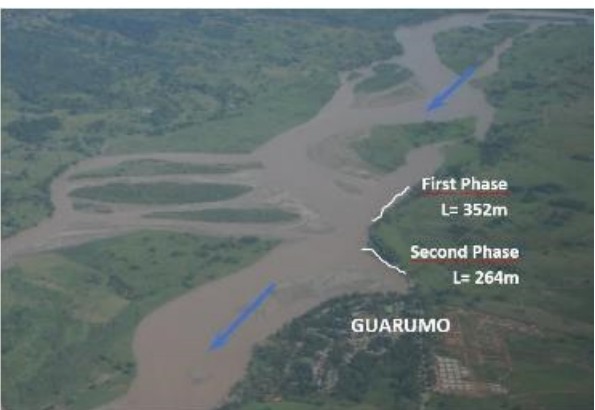

**Figure 17.** Cauca River, Guarumo: location of the two vane fields (Hidroconsulta, 2008).

Figure 18 shows the topography of the Cauca River in Guarumo and the geometry adopted for the design of the vane fields. Initially, the basic hydraulic parameters were determined for a full section flow of 1945 m$^3$/s using the HEC-RAS model (USACE [14]), while the design was made with the BVANE program (Odgaard [10]) for vanes of 2.2 × 6.2 m and 1.2 × 6.2 m, which were used either independently or combined to achieve heights of up to 3.4 m. Thus, the optimal solution was determined for each sector. Figure 19 uses a depiction of subsector 1 as an example of the first stage.

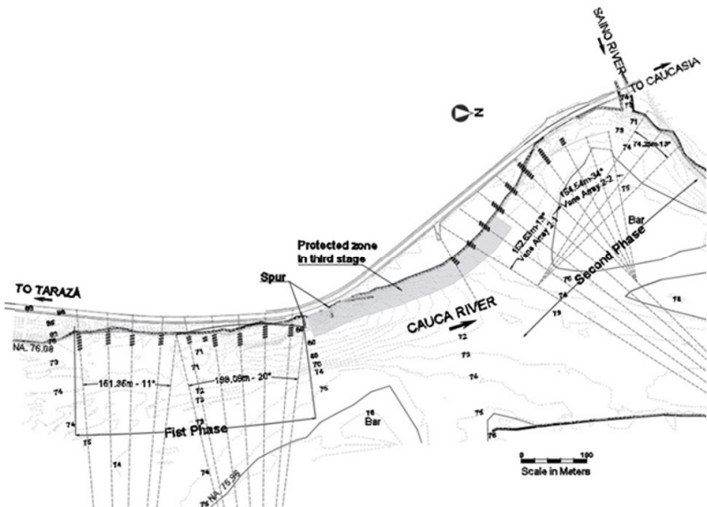

**Figure 18.** Cauca River, Guarumo: topo-batimetry and sector geometry for the design of the submerged vane fields (Hidroconsulta [11]).

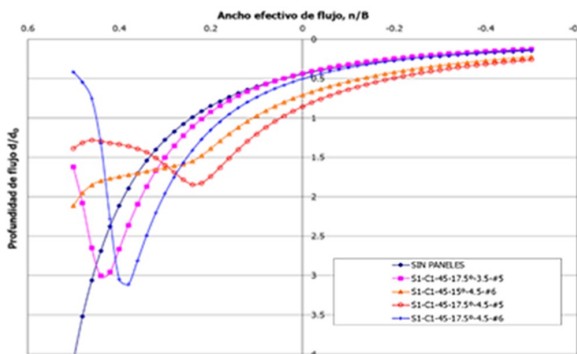

**Figure 19.** Cauca River, Guarumo: results of the BVANE program (Odgaard [10]) for the selection of the optimal solution with vanes using subsector 1, Stage I, as an example.

The field of the first stage consisted of 41 units of vanes, distributed in nine arrays of four, five, or six units each. Each array combined vanes that were 2.2 and 1.2 m tall for heights up to 3.4 m in some cases. The second stage comprised 57 units distributed in eight arrays; in this case, panels of 1.2 × 6.2 m were combined to achieve heights of up to 2.4 m. The panels are duly supported by two metal piles that were 9.0 m in length.

The first stage was completed in 95 days, from 26 December 26 2007 to 31 March 31 2008, while the second stage took 50 days, from 8 September to 25 November 2008 (Figure 20). During the development of the works, there were high flow discharges with velocities of approximately 4.0 m/s as well as changes in the direction of the current that were not present during the initial design of the panels. This presented a frontal attack on the slope between arrays 5 and 6 of the first stage, so the installation of the additional array 5A (consisting of two panels) was required.

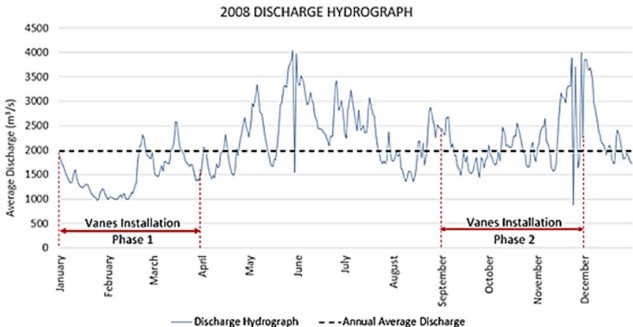

**Figure 20.** Cauca River, Guarumo: series of daily flows in 2008 (Source: La Coquera Station, IDEAM).

The effect of the panel fields installed in Guarumo (Figure 21) became apparent from the very moment of installation, as shown in Figure 22a. Ten years later, in 2019, the protected sector is stable, as shown in Figure 22b.

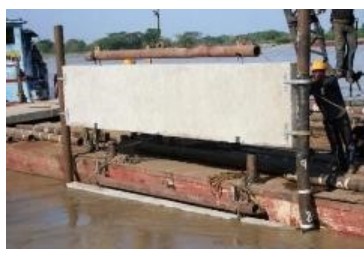 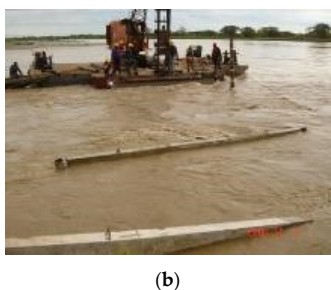

(**a**) (**b**)

**Figure 21.** Cauca River, Guarumo: (**a**) positioning and installation of a double unit; (**b**) a set of vanes in the process of installation (Hidroconsulta, 2008).

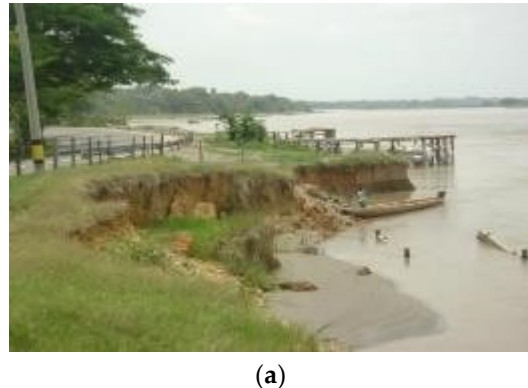 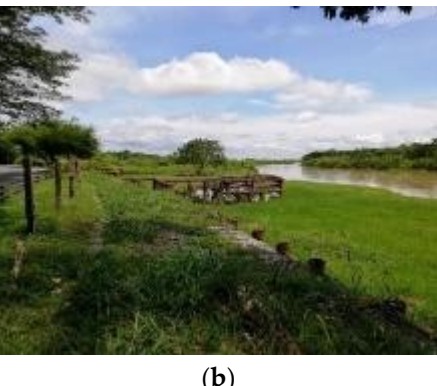

(**a**)                                   (**b**)

**Figure 22.** Cauca River, Guarumo site: (**a**) sedimentation generated shortly after the installation of the assemblies in front of the sector where the channel was initially less than 2 m from the road; (**b**) overview, in 2019, of the effect of vanes around the spurs shown in Figure 15 (Hidroconsulta, 2009; 2019).

*3.5. Ariari River, PR68 Puerto Lleras-Ye de Granada Road, Meta (2018)*

The Granada-Puerto Lleras-Cruce Puerto Rico highway, in the Meta department, runs on the alluvial plain over the left bank of the Ariari River, whose high dynamic threatened the stability of the road towards San José del Guaviare as a result of erosion and undermining that threaten its stability [16] (Project awarded with the 2019 National Engineering Award, of the Colombian Society of Engineers).

Figure 23 shows the condition of the Granada-San José del Guaviare highway in 1994, in the vicinity of the PR68, where the road was slightly moved away from the left bank, to avoid the problem.

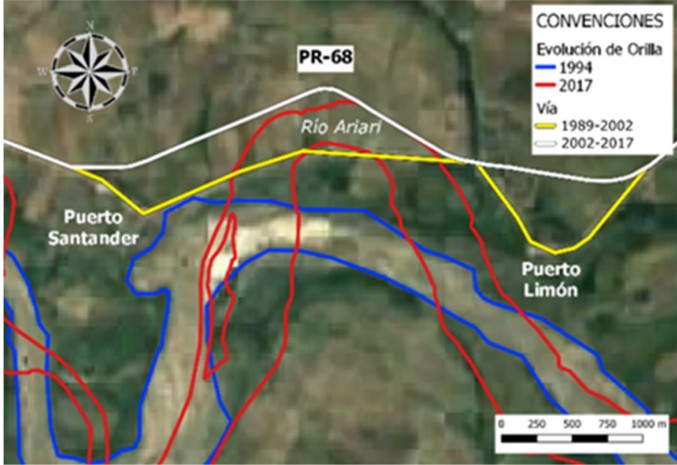

**Figure 23.** Ariari River, PR68: Position of the riverbed in 1994 and 2017, with respect to the original road and northern variant, near the municipality of Fuente de Oro. Note the direct threat along the river over the road (Source: Landsat, 1994 [17] – 2001 [18] – 2002 [19]).

The river continued its process of evolution, with total damage to the road forcing the construction of the northern variant approximately 3 km from the river (Figure 23). However, as a result of the dynamics of the river in the area, this new variant was totally destroyed (Figure 24), revealing an evident need to prioritize control of the evolution of the river around the PR68 over the construction of new variants. Thus, to solve this recurring problem, the use of submerged vane technology was adopted.

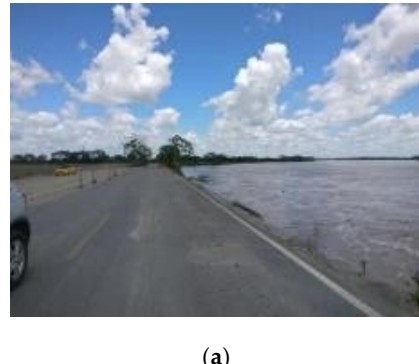
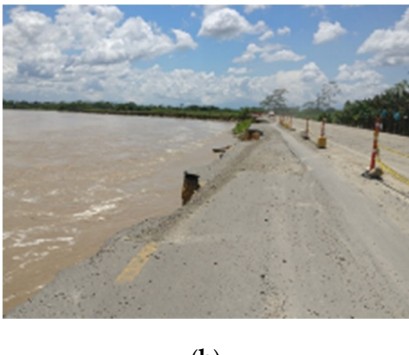

(**a**) (**b**)

**Figure 24.** Ariari River, PR68: (**a**) road sector threatened by the river, without damage (before 2016); (**b**) halftrack bank was destroyed in 2017 (Hidroconsulta, 2017 [16]; 2018 [20]).

The design of the vane system (Hidroconsulta, 2017; 2018) was carried out considering a water discharge of 467 m³/s, with the hydraulic parameters obtained from the HEC-RAS (USACE, 2015 v5.0.1 [21]) and the BVANE model (Odgaard [10]) for panels of 4.4 m in length and 1.6 and 2.2 m in height. The four sectors are shown in Figure 25: one straight that was 420 m long, and three curved sectors of variable lengths and radii, all of which combined to determine a total length of 1110 m along the riverbank. As a result, the vane field was composed of 137 units distributed in 31 arrays, as shown in Table 3. Figure 26 shows the results of the VBANE model.

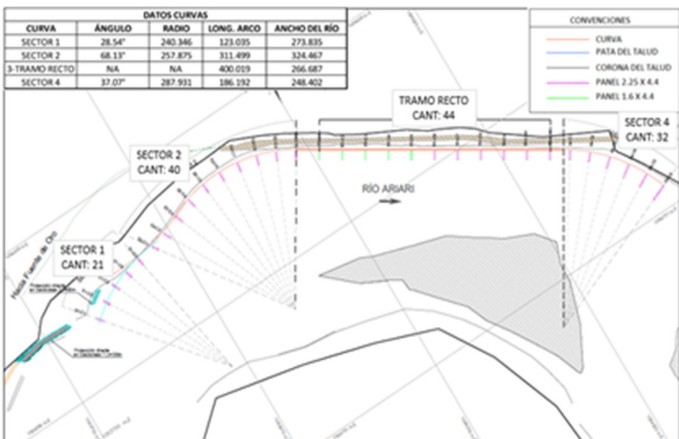

**Figure 25.** Ariari River, PR68: sectors defined for the design and layout of the vane system (Hidroconsulta [16]).

**Table 3.** Ariari River, PR68: optimal design of the initial version of the vane system.

| Sector | Units per Array | Separation (m) | | Protection Length (m) | Number of Arrays | Total Vanes |
|---|---|---|---|---|---|---|
| | | Between Vanes | Between Arrays | | | |
| 1 | 4 o 5 | 3.5 | 30 | 150 | 5 | 21 |
| 2 | 5 | 4 | 35 | 350 | 9 | 40 |
| 3A* | 4 | 4 | 40 | 170 | 5 | 20 |
| 3B | 4 | 4 | 40 | 265 | 6 | 24 |
| 4 | 4, 5, 6 and 7 | 4 | 35 | 175 | 6 | 32 |
| | | | TOTAL | 1.110 | 31 | 137 |

*1.6 × 4.4 m vanes.

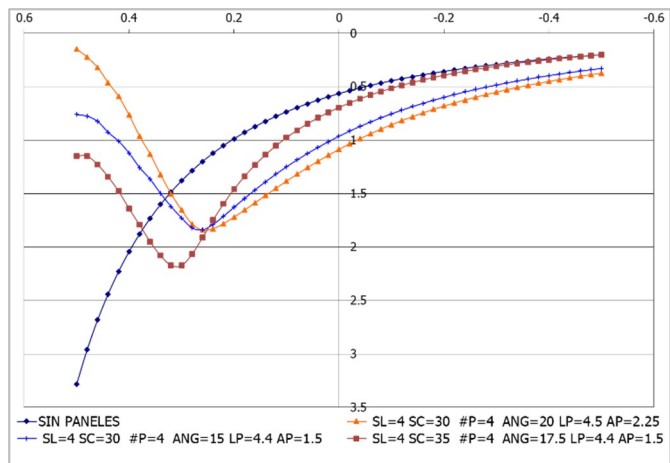

**Figure 26.** Ariari River, PR68: hydraulic modeling results with BVANE for different combinations of elements for the protective design, in a representative section.

Seven months after the design was completed, beginning 4 May 2018, the C3 to C8 arrays were installed, in order, followed by the C1 to C3 assemblies. At this point, unfortunately, the rainy period intensified in the region, the intensities and durations of which generated increasingly frequent, large water flows and high transports of solids and vegetal materials. Thus, a wide isle of sediment was developed in the center of the channel, upstream of the arrays already installed. Fractionation of flows in the sector directed them from the right arm to the left bank, thus exacerbating the erosion of this bank. In addition, an effect of this significant new evolution was the even greater deterioration of the remaining embankment of the road, which in practice was completely razed (Figure 27).

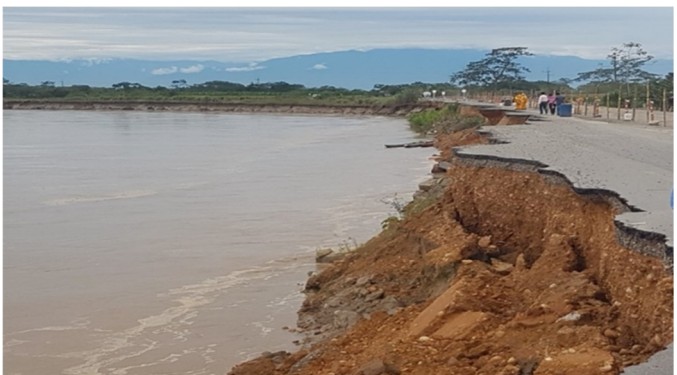

**Figure 27.** Ariari River, PR68: increased damage to the road embankment from intensive rains during May and June 2018 (Source: Hidroconsulta).

As a result, it became inevitable to redesign the vane system and radically counteract the frontal attack of flows on the bank. Thus, version 2 of the system was developed (Figure 28), with the addition of 34 units. Arrays C-00 and C-01 were placed on the major thalweg, and arrays C-0, C-1, and C-2 were placed in front of the upstream end of the system; all were installed between 9 and 20 June 2018. However, two sites were identified during the installation that showed a potential risk of erosive attacks by flows, so, to prevent possible future problems, version 2 of the redesign was complemented with 15 vanes that distributed in three new intermediate arrays: C18, C19, and C20. Thus, the vane field was eventually made up of a total of 186 units (Figure 28) that were distributed in 39 arrays.

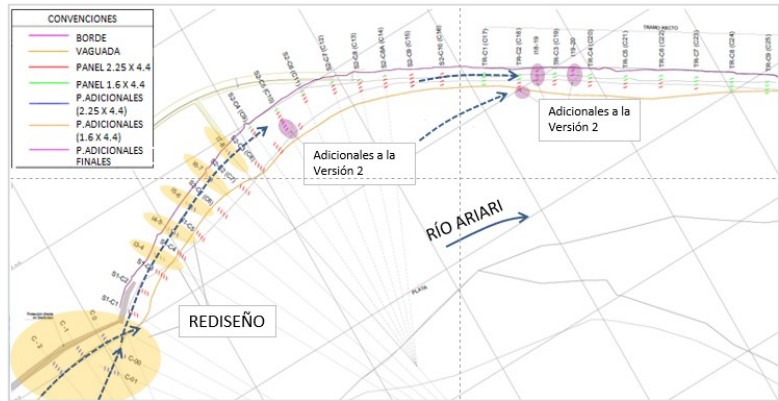

**Figure 28.** Ariari River, PR68: vane field installed according to version 2 of the redesign and additional vanes.

For the system, flat plates in reinforced concrete class C (280 kg/cm$^2$), which were prefabricated in situ and lightened with 0.22 m thick polystyrene sheets, were used (Figure 29). To fix the vanes, steel pipe piles with 7" inner diameters and minimum wall thicknesses of 6 mm (0.24") were installed. The installation process is shown in Figure 30.

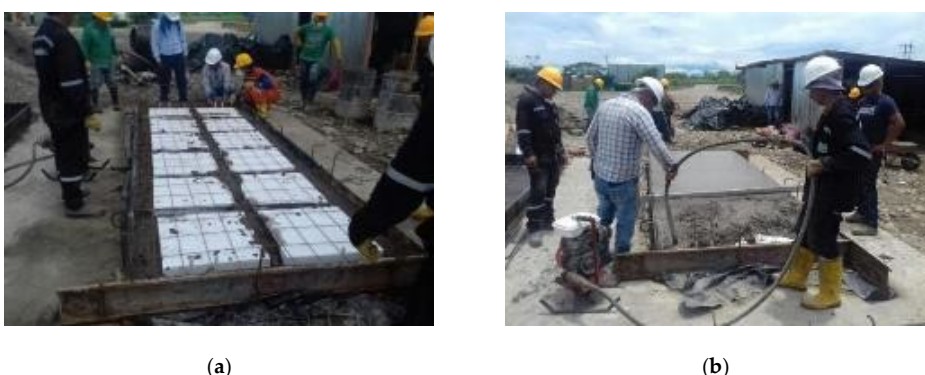

(**a**)                                                    (**b**)

**Figure 29.** Ariari River, PR68: on-site manufacturing of reinforced concrete vanes, lightened. (**a**) Provision of the lightening material; (**b**) placement of concrete.

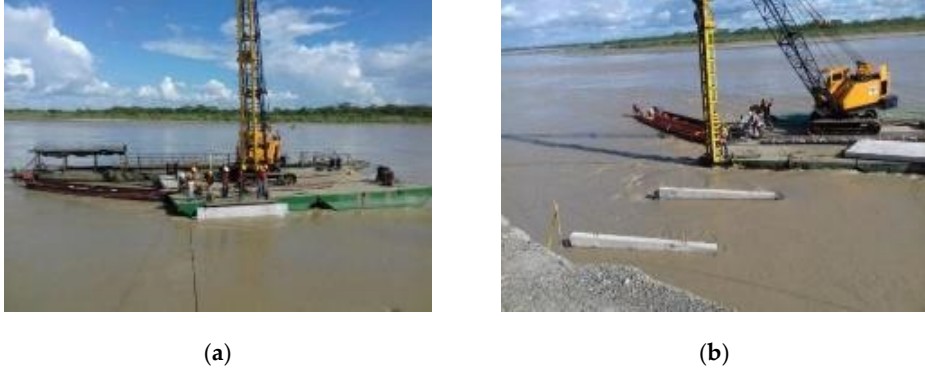

(**a**)                                                    (**b**)

**Figure 30.** Ariari River, PR68: installation of vanes. (**a**) Positioning according to the design; (**b**) array in the process of installation.

In the short period since construction between May and September 2018, the beginning of the sedimentation process envisaged in the designs was evident (Figures 31 and 32), and continued over time. The cost of the project amounted to only USD 1 million, which is very favorable compared to the construction of variants that have an average value from USD 1.5 million to USD 3.0 million per

kilometer. Undoubtedly, the use of submerged vane technology is highly favorable due to the savings it generates, among other benefits.

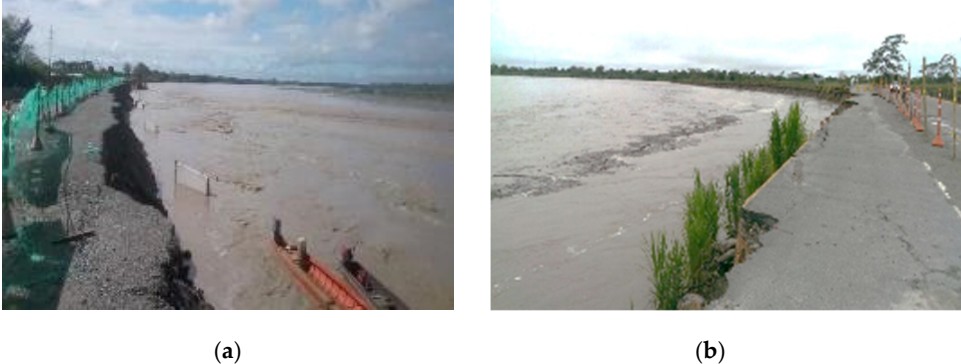

|  |  |
|:-:|:-:|
| (**a**) | (**b**) |

**Figure 31.** Ariari River, PR68: surface condition of the flows shows the transfer of the thalweg to the center of the riverbed by the panels. (**a**) Downstream view; (**b**) upstream, facing the semi-destroyed road. Compare with Figure 24 (Source: Hidroconsulta).

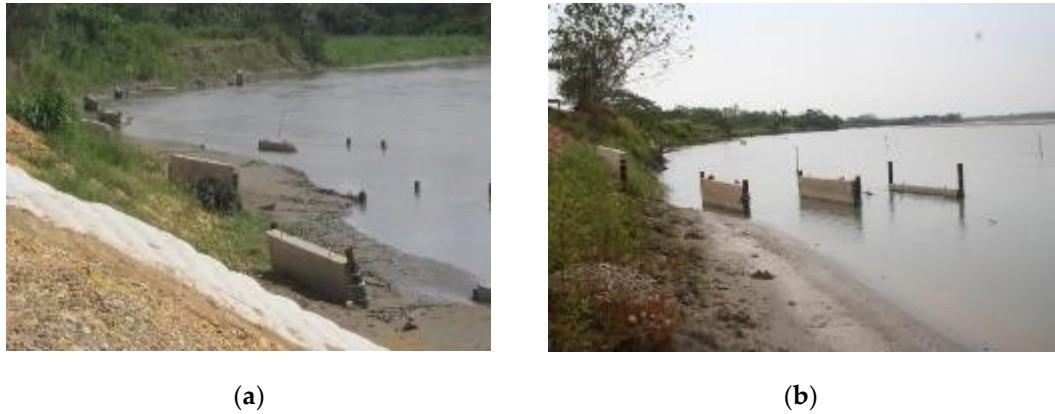

|  |  |
|:-:|:-:|
| (**a**) | (**b**) |

**Figure 32.** Ariari River, PR68: sedimentation in process around the vanes. (**a**) In the upstream sector; (**b**) before the straight section (Source: Hidroconsulta).

## 4. Conclusions

The success of the submerged vane technology in Colombia is demonstrated by the five representative projects contained in this article. The designs and recommendations in the technical articles that support them were rigorously applied, with variations introduced when required by local conditions.

The models were adapted to the conditions of Colombian rivers with medium flows, which exceed those of projects of this type carried out worldwide (Odgaard [6]). In addition, procedures for the manufacture and installation of the vanes were developed and improved continuously, based on the conditions of each project, location, diversity of hydraulic conditions and availability of materials and equipment in the areas, among other factors. In Colombia, these projects use thin plates and prefabricated solid or lightened concrete or metal lattice coated with high-density polypropylene sheets.

The experiences described herein and similar projects in Colombia demonstrate the successful application of the models developed by the Institute of Hydraulic Research of the University of Iowa for the formulation of solutions to problems related to the evolution of rivers and the processes of erosion and undermining using submerged vanes. These projects reaffirm the effectiveness of the technology and the realization of significant economic and environmental benefits over traditional solutions.

Through a sustained effort over more than 25 years, more efficient use has been made of public resources destined to solve the problems of erosion, undermining, sedimentation, and improvement of

navigation, providing an important service to the country and achieving significant savings on each project. It is significant to note that the application of this new technology in Colombia is done with local engineering, materials and labor.

Table 4 provides a summary overview of the five projects and supports the next discussion of the final results.

**Table 4.** Overview of the projects described.

| Project (Year) | Aromáticos (1993) | Trementino (2005) | Puerto López, Sector 2 (2007) | Guarumo: of the Steps (2009) | PR68 (2018) |
|---|---|---|---|---|---|
| RIVER | Magdalena | Sinú | Metica | Cauca | Ariari |
| $Q_{DESIGN}$, m$^3$/s | 3850.0 | 645.0 | 1132.0 | 1945.0 | 467.0 |
| Total, arrays | 9 | 13 | 27 | 17 (9 + 8) | 39 |
| Total, Vanes | 71* | 45** | 134*** | 98 (41 + 57) | 186 |
| Protected length, m | N/A | 630.0 | 1050.0 | 616.0 | 1110.0 |
| Starting date | 01/06/93 | 13/09/04 | 01/29/2007 | 26/12/07 08/09/08 | 04/05/18 |
| Ending date | 23/08/93 | 20/12/04 | 28/07/2007 | 31/03/08 25/11/08 | 10/08/18 |
| Savings obtained | USD 1.15 million/ year for more than ten years. | USD 1.6 million | USD 1.9 million | USD 1.07 million plus the unquantified cost of dredging | USD 5 to 10 million [1] |

\* Reinforced concrete vanes. ** Vanes in metal lattice, coated with high-density polypropylene sheet. *** Reinforced concrete vanes lightened with expanded polystyrene sheets. [1] Construction cost of a 3.0 km variant.

The results obtained with the use of the submerged vane technology for sediment control and riverbank protection in Colombia, as described in each of the five projects included in this paper, show favorable effects on the environment, as well as reduced costs of the solution itself and of maintenance of the systems.

Environmental impact: Submerged vane systems require far lower volumes of material than most other types of solutions, so environmental demand on mineral resources is minimal. Being that the panels are not usually visible from the surface in a submerged condition, they do not impact the landscape or affect the navigation nearby.

There have been no studies on possible impacts on natural conditions of the aquatic environment that may affect the fauna and flora of the rivers where the panels have been installed, but no complaints have been received from the inhabitants about qualitative or quantitative changes of these species after the panels were installed.

Economic impact: The cost of submerged vane solutions has been much lower than any other solutions that had been designed for sites for which the final decision was to adopt the submerged vane solution. Information on the economic benefits obtained through the five projects included is presented in Table 4.

Maintenance costs: None of the projects built have required maintenance work for proper operation or preservation since the date of their construction, which is not the case with the other solutions that have been used to achieve the same effect.

No floating material trapping occurs, as the vanes work at the depth of the bed allowing those elements pass over. As observed, even in the cases of panels near the shores—which operate at a lower depth or even protrude from the surface—there has been no accumulation of material that could favor a section reduction and cause flooding.

In conclusion, the improved economic, environmental, and maintenance results show the favorable possibilities in implementing the vane technologies on large rivers such as in Colombia. Further research and dissemination of results such as those explained here are the logical next steps.

**Author Contributions:** Conceptualization, formal analysis, investigation and methodology of all projects, C.R.-A.; Formal analysis, Methodology for projects 3.3, 3.4 and 3.5, S.D.-M.; Construction supervision of project 3.1, C.R.-A.; Planning, programming and construction supervision of projects 3.2 to 3.5, A.D.-A.; Writing original draft, C.R.-A.; writing review and editing, C.R.-A., S D.-M. and A.D.-A. All authors have read and agreed to the published version of the manuscript.

**Funding:** The studies, designs and construction of the special hydraulic works supporting this document were financed by the Colombian Petroleum Company ECOPETROL (Contract: 2-35110-389-292556, Magdalena River, Aromatic); and by the Instituto Nacional de Vías INVIAS (Contract 314/2005, Sinú River, Trementino; Contracts 314/2005 and 2716/2006, Metica River, Puerto López; Contracts 3138 of 2007 and 0493 of 2008, Cauca River, Guarumo; Contract 00943 of 2017, Ariari River, PR68).

**Acknowledgments:** ECOPETROL and INVIAS are thanked for the collaboration and authorization to publish the results of these works. In a special way, the support of the engineers and collaborating specialists, and the technical and auxiliary staff of HIDROCONSULTA, is recognized for the execution of office and field work, and in the development of the designs and construction of the works.

**Conflicts of Interest:** The authors declare no conflict of interest.

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
