# Peer review of "Submerged Vane Technology in Colombia: Five Representative Projects"

_water, doi:10.3390/w12040984_

Round 1

Reviewer 1 Report

This paper provides a review of 5 consulting projects that were carried out in Colombia. Although the technology installed is relevant to sediment transport and has scientific implications, this paper appears to highlight specific built construction projects rather than provide a hypothesis, methods, analysis, and results. 

This paper would require substantial revision in order to be suitable for this journal. 

The first paragraph does not indicate what the paper is actually investigating. There are many very general statements made throughout this paper rather than focusing specifically on what was the analysis completed by the authors.

The paper also overstates in a subjective manner the determination of whether or not a project was successful (Line 39 "remarkable success"; Line 38 advertising the specific name of a consulting company; Line 88 "of great importance". By doing so, the authors seem to be advertising projects rather than presenting a scientific analysis. 

Author Response

Point 1.  This paper provides a review of 5 consulting projects that were carried out in Colombia. Although the technology installed is relevant to sediment transport and has scientific implications, this paper appears to highlight specific built construction projects rather than provide a hypothesis, methods, analysis, and results. 

This paper would require substantial revision in order to be suitable for this journal. 

Response 1. Thank you for this opinion.  We have prepared a substantial revision.

Point 2. The first paragraph does not indicate what the paper is actually investigating. There are many very general statements made throughout this paper rather than focusing specifically on what was the analysis completed by the authors.

Response 2.  We added a more specific research question in the introduction and have worked through the document to identify and unsupported general statements so that we can provide a focus and relate them to the research question.

Point 3. The paper also overstates in a subjective manner the determination of whether or not a project was successful (Line 39 "remarkable success"; Line 38 advertising the specific name of a consulting company; Line 88 "of great importance". By doing so, the authors seem to be advertising projects rather than presenting a scientific analysis.

Response 3. We have screened the paper to remove any reference to commercial work and have clarified the measures of success according to the research question.

Reviewer 2 Report

The paper reports on a study on effects of submerged vane technology on sediment control and erosion of the banks and development and migration of meandering, which are one of the most important problems of river management. Therefore the subject is very important. I find the paper and the presented solution very interesting. The manuscript was well structured and easy to read. However, in my opinion in current form the article, although very interesting, seems to be a report, the scientific aspect is insufficient. It should be changed before publishing. My suggestions:

The introduction should contain more information concerning the disadvantages of mentioned by the authors traditional solutions used in the riverbeds to decrease water erosion processes. The aim of the paper should be reworded to make it more scientific. “Publicize the positive results…” (Line 44) in itself can’t be the aim. I would suggest I would for example: “Assessment of submerged vanes effects on the riverbed stabilization in cross-section on the example of five case studies” To strengthen the scientific character of the article, I would suggest adding a “Discussion” at the end. In the discussion, Authors should refer to: environmental aspects, which are completely omitted, and yet it is extremely important. Each and every form of interference of a river bed has an effect on its biocoenosis. Compared to traditional bank reinforcements, the shoulder blades cover a larger area of ​​the riverbed, which significantly limits the development of plants, and thus the number of habitats. What is the impact on aquatic submerged flora and fauna and physico-chemical properties of water (lighting, temperature, dissolved oxygen). risk of flooding - do the submerged vanes not increase the risk of flooding (e.g. Stopping dead wood can lead to blockages). utilitarianism of submerged vanes, because there is also a lack of a broader description of the technical possibilities of using this solution (is it a universal solution or are there any restrictions, as e.g. minimum width or depth of the river), as well as the principles of vanes exploitation (e.g. do you need to maintain it, how often, what is the cost?).

I hope the authors are able to put at least some of this additional information. This will significantly increase value of the paper.

Author Response

Point 1. The paper reports on a study on effects of submerged vane technology on sediment control and erosion of the banks and development and migration of meandering, which are one of the most important problems of river management. Therefore the subject is very important. I find the paper and the presented solution very interesting. The manuscript was well structured and easy to read. However, in my opinion in current form the article, although very interesting, seems to be a report, the scientific aspect is insufficient. It should be changed before publishing.

Response 1. Thank you for your positive comments.  We have added a specific research question in the introduction and have worked to remove the impression that the paper is a report, rather than a scientific investigation.

My suggestions:

Point 2. The introduction should contain more information concerning the disadvantages of mentioned by the authors traditional solutions used in the riverbeds to decrease water erosion processes.

Response 2. We have added information about these disadvantages, see the track changes copy.

Point 3. The aim of the paper should be reworded to make it more scientific. “Publicize the positive results…” (Line 44) in itself can’t be the aim. I would suggest I would for example: “Assessment of submerged vanes effects on the riverbed stabilization in cross-section on the example of five case studies”

Response 3. Thank you for this positive suggestion, which we have considered in reformulation of the research question.

Point 4. To strengthen the scientific character of the article, I would suggest adding a “Discussion” at the end. In the discussion, Authors should refer to: environmental aspects, which are completely omitted, and yet it is extremely important. Each and every form of interference of a river bed has an effect on its biocoenosis. Compared to traditional bank reinforcements, the shoulder blades cover a larger area of ​​the riverbed, which significantly limits the development of plants, and thus the number of habitats. What is the impact on aquatic submerged flora and fauna and physico-chemical properties of water (lighting, temperature, dissolved oxygen). risk of flooding - do the submerged vanes not increase the risk of flooding (e.g. Stopping dead wood can lead to blockages).

Response 4. Thank you for this helpful discussion of how we can strengthen the scientific aspects of the paper.  We have added material on the environmental questions you mention, see the track changes version.

Point 5. utilitarianism of submerged vanes, because there is also a lack of a broader description of the technical possibilities of using this solution (is it a universal solution or are there any restrictions, as e.g. minimum width or depth of the river), as well as the principles of vanes exploitation (e.g. do you need to maintain it, how often, what is the cost?).

Response 5. We have added in the final discussion a description of the technical possibilities.

Point 6. I hope the authors are able to put at least some of this additional information. This will significantly increase value of the paper.

Response 6. We have incorporated all of Reviewer 2’s suggestions.

Round 2

Reviewer 2 Report

The paper reports on a study on effects of submerged vane technology on sediment control and erosion of the banks and development and migration of meandering, which are one of the most important problems of river management. The subject is very important. I find the paper interesting, however, in previous version I have noticed some inaccuracies.

In the new version the authors referred to all my comments. The paper improved considerably. The introduction was rewritten, the environmental and economic aspects was added. The technical possibilities was clarified in detail. I think, now the paper is suitable for publication in the Water.

As a side note, I would like to suggest to the authors an attempt to assess the environmental impact of submerged vanes. In Europe it is a very important aspect. In the light of the Water Framework Directive (2000/60/EC), the management of surface water must not contribute to the deterioration of its quality. If it is confirmed that this impact is insignificant, submerged vanes can be a very good compromise between the interests of water users and the environmental objectives of the rivers.